# A Review on the Role of Stem Cells against SARS-CoV-2 in Children and Pregnant Women

**DOI:** 10.3390/ijms222111787

**Published:** 2021-10-30

**Authors:** Fatemeh Sanie-Jahromi, Yaser NejatyJahromy, Rahim Raoofi Jahromi

**Affiliations:** 1Poostchi Ophthalmology Research Center, Shiraz University of Medical Sciences, Shiraz 7134997446, Iran; fsanie@sums.ac.ir; 2Institut für Physikalische und Theoretische Chemie, Rheinische Friedrich-Wilhelms-Universität Bonn, 53012 Bonn, Germany; 3Department of Infectious Disease, Peymanieh Hospital, Jahrom University of Medical Science, Jahrom 7414846199, Iran

**Keywords:** COVID-19, quiescent stem cell, active stem cells, cell cycle, children, pregnant women, ACE/ACE2

## Abstract

Since the COVID-19 outbreak was acknowledged by the WHO on 30 January 2020, much research has been conducted to unveil various features of the responsible SARS-CoV-2 virus. Different rates of contagion in adults, children, and pregnant women may guide us to understand the underlying infection conditions of COVID-19. In this study, we first provide a review of recent reports of COVID-19 clinical outcomes in children and pregnant women. We then suggest a mechanism that explains the curious case of COVID-19 in children/pregnant women. The unique stem cell molecular signature, as well as the very low expression of angiotensin-converting enzyme 2 and the lower ACE/ACE2 ratio in stem cells of children/pregnant women compared to adults might be the cause of milder symptoms of COVID-19 in them. This study provides the main molecular keys on how stem cells can function properly and exert their immunomodulatory and regenerative effects in COVID-19-infected children/pregnant women, while failing to replicate their role in adults. This can lay the groundwork for both predicting the pattern of spread and severity of the symptoms in a population and designing novel stem cell-based treatment and prevention strategies for COVID-19.

## 1. Introduction

Coronavirus disease of 2019 (COVID-19) outbreak has been a popular subject of research since its emergence in December 2019 in Wuhan, China. The severe acute respiratory syndrome coronavirus 2 (SARS-CoV-2) spread rapidly, and the disease it causes was declared by the World Health Organization (WHO) an international health emergency on 30 January 2020 [1]. The spread rate of this virus is extremely high, and the reported mortality rate is variable in different regions [2]. Infected patients show a spectrum of symptoms, ranging from asymptomatic infections to acute respiratory distress syndrome (ARDS) [3]. SARS-CoV-2, the etiologic agent responsible for COVID-19, is a virion with single-stranded RNA and belongs to the beta-coronavirus (CoV) genus. To enter the host cells, the virus utilizes its spike glycoprotein (S protein) to recognize the angiotensin-converting enzyme-2 (ACE-2) receptor on the host and the transmembrane serine protease 2 (TMPRSS2), a priming enzyme required for processing the S proteins of SARS-CoV-2 [4,5]. It seems that the first host cells for SARS-CoV-2 are the ciliated cells of the airway and then the type-2 alveolar (AT-2) epithelial cells of the lung [6]. However, the cells of other tissues, particularly, nasal, intestinal, and corneal cells, are also shown to be easily targeted by SARS-CoV-2, which could be attributed to the high expression of ACE2 and TMPRSS2 on these cells [7]. Virus-infected cells produce a wide range of cytokines, including interferons β, γ, and CXCL10 (C-X-C motif chemokine 10), igniting a cytokine storm [8,9]. About 80% of patients will develop a mild disease at this stage, which is self-controlled and does not require hospitalization or intensive medical care [10]. Unfortunately, the disease progresses to a very severe condition in about 20% of the patients (including severe lung alveolar cell damage, excessive inflammation, and difficulty in breathing and oxygen delivery). Antiviral and anti-inflammatory therapies along with supportive care are the main treatments for COVID-19 [11]. Additionally, biological interventions such as mesenchymal stem cell (MSC) therapy have found a special place in COVID-19 therapeutic research [12]. MSCs are undifferentiated multipotent cells and are the most ubiquitous types of adult stem cells. After birth, MSCs are found in various tissues including bone, fat, cartilage, umbilical cord, skin, teeth, and many other tissues. MSCs function to grow, repair, and replace damaged cells and heal injuries [13]. MSCs have unique properties that make them a promising candidate for cell therapy of COVID-19 [14]. These cells can secrete a wide range of immunomodulatory cytokines and growth factors, such as keratinocyte growth factor (KGF), angiopoietin-1 (Ang1), hepatic growth factor (HGF), nitric oxide (NO), transforming growth factor-*β* (TGF-*β*), prostaglandin E2 (PGE2), and indoleamine2,3-dioxygenase (IDO) [14,15,16,17]. In this way, MSCs can prevent ARDS-induced apoptosis of alveolar epithelial and endothelial cells, inhibit SARS-CoV-19-associated inflammation, and regenerate the damaged lung tissue [15,16,17,18,19,20]. MSCs interestingly express the major histocompatibility complex (MHC) at a very low level, which makes these stem cells non-immunogenic in host patients [21]. Moreover, the ACE2 receptor and the TMPRSS2 enzyme are also expressed in MSCs at very low levels, and previous studies indicated that the SARS-CoV-2 virus cannot infect these cells [22,23,24]. Positive therapeutic results have been reported when utilizing MSCs in the treatment of respiratory complications [25,26,27,28,29]. MSCs are thought to use both juxtacrine (i.e., direct cell–cell interaction) and paracrine (secretion of extracellular vesicles and exosomes) paths [30]. MSCs help regenerate tissues and reduce cell-based and humoral immune responses through immunomodulating factors such as FGF, HGF, EGF, VEGF [31]. For instance, secretion of HFG by MSCs helps to induce dendritic cell immune tolerance and ease acute lung injury (ALI) [32]. MSCs are shown to reduce the inflammation of epithelial lung cells by transporting miR21-5p and increasing the polarization of alveolar macrophages from the pro-inflammatory to the anti-inflammatory phenotype [33]. Pro-inflammatory factors produced by M1 macrophages at the site of injury activate resting MSCs, whose induced immunosuppressive characteristics, in turn, lower the overall immune response [34]. MSCs further suppress the proliferation of lung Th1 and Th17 helper T lymphocytes through a plethora of secreted cytokines including TGF-β and HGF [35]. MSCs similarly suppress the cell cycle of B lymphocytes, resulting in a diminished expression of a series of chemokine receptors (such as CXCR4 and CXCR5) and immunoglobulins (IgM, IgG, IgA) [36]. Alongside the anti-inflammatory actions of MSCs, they further facilitate the restoration of tissues damaged by ARDS, ALI, and other repertory inflammatory complications through their angiomodulation [37]. Their support for the generation of blood vessels, through the production of a spectrum of proangiogenic factors such as VEGF, HGF, TGF-β, MCP-1, angiopoietin-1, etc., reinforces the recovery process of injured pulmonary tissues by addressing the oxygen supply [38]. Therapeutic applications of MSCs have been considered in several autoimmune disorders in the last decade, and the effectiveness and safety of this treatment have been proven [39]. Recently, Afarid et al. discussed clinical trials evaluating MSCs application for the treatment of COVID-19 infection, demonstrating the unique features of stem cells in the immunomodulation of the COVID-19-induced cytokine storm, the regeneration of alveolar epithelial and endothelial cells, and the secretion of factors that could prevent lung fibrosis and severe COVID-19 manifestations [14]. Given the above, along with the universal presence of MSCs in the human organism, from infants to the elderly, it is not farfetched to expect an important protective role of MSCs against COVID-19. However, a look at the COVID-19 prevalence, severity, and mortality rate in relation to different age brackets shows that MSCs probably act differently in different age groups of the population.

The mortality rate of COVID-19 is about 2% and is strongly correlated to the age of patients [10]. The average age of patients confirmed as positive for COVID-19 is 49–56 years [40]. It seems that the susceptibility to the virus increases with age. Although cases of infection were reported in children, clinical observations indicate that COVID-19 manifests milder symptoms in the youth compared to adults [41,42]. In a study by Hun et al., 53 COVID-19 patients from different age groups were compared for their clinical manifestations. The study revealed that patients older than 40 years showed significantly more symptoms associated with disease exacerbation, including CT abnormalities, multiple lung lesions, and bilateral lung involvement. In addition, this group needed to receive more intensive antiviral drug regimens to control the disease [43]. Interestingly, the study of COVID-19 in pregnant women has shown that, similar to the youth, pregnant women have a lower infection rate and show milder symptoms of COVID-19. In the present study, we reviewed the reports of COVID-19 in children and pregnant women. In relation to the lower incidence and severity of the disease in pregnant women and children, we highlight a common feature of these two groups, namely, the stem cell characteristics, that may lead to resistance to this disease. Finally, we suggest a significant role for actively proliferating stem cells in making pregnant women and children more tolerant to COVID-19.

## 2. COVID-19 in Children

Pediatric COVID-19 has been investigated in several studies around the world. A survey by the China Centers for Disease Control and Prevention on 72,314 COVID-19 patients showed that children below 10 years of age account for less than 1% of COVID-19 cases [44]. In another study in China, 1391 children (mean age of 6.7 years) were assessed, from which 171 cases were confirmed to be COVID-19-positive. This survey reported a range of disease exacerbation in children with COVID-19 from asymptomatic to severe cases (27 patients with symptomatic infection, 33 patients with upper respiratory tract infection, 111 pneumonia patients). Just 3 of 171 patients (all 3 with coexisting conditions) required intensive medical care, and one death was reported [45]. This study showed that, unlike infected adults, most infected children manifest a milder clinical course; a large fraction of asymptomatic infections was also observed. Similar results on the prevalence and severity of pediatric COVID-19 were reported in a study of the early onslaught of COVID-19 in the United States. In this study, out of 149,082 COVID-19-positive cases, 2572 (1.7%) were children under 18 years of age The researchers also observed that, among these pediatric cases, infants under one year of age (particularly those with coexisting conditions) were the largest group in need of hospitalization [46]. Similar results on the lower rate of infection and mortality in COVID-19-infected children have been reported by other medical centers [47,48,49]. In a review of COVID-19-positive cases of children and infants in China, the USA, Korea, and Taiwan, Jeng et al. indicated that, although the incidence of SARS-CoV-2 infection is relatively low in children and their symptoms are less severe than those of adults, critical conditions were reported in some cases, particularly for those with preexisting conditions [50]. Overall, COVID-19 mortality is rare in children, and clinical intervention is needed almost exclusively in children with coexisting conditions [45,46]. Table 1 shows a summary of reports on COVID-19 in children from different medical agencies (see references [10,45,46,47,51,52,53,54,55,56,57,58,59,60]). Hence, published studies have consistently shown that, in children, COVID-19 patients manifest milder symptoms, need less frequent hospitalization, and show a lower fatality rate.

## 3. COVID-19 in Pregnant Women

The early studies of COVID-19 cases suggested that pregnant women did not exhibit a higher susceptibility to infection with the novel coronavirus [40]. Chen et al. reported nine cases of COVID-19 in pregnant women, all of whom were in their third trimester. Their clinical features were the same as those of non-pregnant women, while none of the pregnant patients in this study required mechanical ventilation [61]. Della Gatta et al. examined 51 pregnant patients and suggested that COVID-19 prognosis for both mothers and neonates is more promising, when compared to that for the general public [62]. In a report by a WHO–China joint mission on COVID-19, an analysis of 147 pregnant women showed that pregnancy does not impose a higher risk of developing severe pneumonia due to COVID-19 [63]. Dashraath et al. reported a 0% mortality of COVID-19-infected mothers and asserted more encouraging COVID-19 outcomes for pregnant women in comparison with MERS and SARS [64]. Pierce-Williams et al. also described the clinical course of 64 COVID-19-infected pregnant women with no maternal mortality [65]. Hantoushzadeh et al., on the other hand, documented seven deaths out of nine severe cases of COVID-19-infected pregnant women [66]. Although common symptoms were observed in the COVID-19-positive pregnant women, the milder symptoms and few reported deaths in pregnant women is a significant observation [67]. These reports indicate that pregnant women are less susceptible to COVID-19 and can battle this infectious disease better than other groups. Early studies do not indicate that there is a drastic difference between the levels of infection resistance in pregnant women in different trimesters; however, more experimental data might be needed to confidently draw a conclusion on the subject [68,69,70]. The intrinsic immune tolerance of pregnancy in general, and all or part of its underlining molecular mechanisms, including the roles of hormones, can contribute to the curious resistance of pregnant women to COVID-19 [71,72,73]. Alongside reviewing COVID in children and pregnant women, the authors tried to highlight common factors between these two demographic groups which may explain COVID-19 tolerance in these populations.

## 4. What Makes Children and Pregnant Women More Tolerant to COVID-19?

Although children and pregnant women are usually considered to be high-risk populations for infections [74,75], they show remarkable resistance in the COVID-19 pandemic. The statistics of those suffering from the disease around the world demonstrate that COVID-19 takes most of its victims from the elderly population [76,77]. Noteworthy is that the immune system is commonly weaker in children and it is modulated during pregnancy, as a result of which, children and pregnant women are more susceptible to contagious diseases [78,79]. Given the above, the question that naturally arises is “what increases the resistance of pregnant women and children to this dangerous viral disease?”. In terms of cell biology, one of the common features of these two groups is their stem cell composition. Stem cells in children/pregnant women differ in many molecular features from their counterparts in the elderly. Here, we point out two main differences between stem cells in the body of children/pregnant women and those of adults, namely, “active versus quiescent stem cells” and the “ACE/ACE2 ratio”. We hypothesize that these two differences in stem cell composition are associated with the observed increased resistance to COVID-19.

### 4.1. Active Versus Quiescent Stem Cells

Stem cells in adults (including hematopoietic stem cells, muscle satellite cells, and MSCs) are mostly quiescent, in the G0 phase of the cell cycle, with a lower rate of proliferation and a distinct epigenetic and molecular signature [80,81]. In a recent study by Liu et al., adipose-derived mesenchymal stem cells (hASCs) were evaluated for a range of molecular and cellular characteristics of stemness in different age groups [82]. It was shown that aging reduces both the proliferation rate of hASCs and their potential for adipogenic and osteogenic differentiation. Mitochondrial-specific reactive oxygen species (ROS) and p21 expression were also shown to be significantly increased with age. This study also revealed that stem cells from the older population have an impaired ability of migration and decreased expression of chemokine receptors, such as CXCR4, compared to those of the youth and children [82]. CXCR4 is actively involved in cell proliferation and tissue regeneration and has a major role in G0/G1 transition [83]. Overall, the results of Liu’ study showed that although hASCs derived from different age groups are phenotypically similar, they differ significantly in function [82]. That is, children have more active stem cells in their bodies. Similar studies demonstrated that the regenerative properties of adult stem cells decrease with age, and the molecular pathways in these cells change, resulting in diminished potential for efficient tissue repair [84,85,86,87].

Similar to children, pregnant women have active fetal-derived stem cells circulating in their bodies, that are able to combat illnesses and repair maternal injured tissues [88]. Fetal stem cells are multipotent stem cells derived from fetal blood and tissues. These cells are more limited in growth potential than pluripotent embryonic stem cells [89], although their proliferation rate and regenerative properties are higher than those of MSCs from adults [90,91,92]. Fetal-derived stem cells start circulating into the maternal bloodstream in the sixth week of gestation and increase in number by gestational age. It is estimated that 1–6 cells/mL of fetal cells are present in the maternal venous blood in the second trimester of pregnancy. Fetal cells decrease after delivery; however, a small fraction of fetal cells remain in the maternal blood for decades after delivery [88]. The fetal cells that transfer through the placenta are from specific cell types, including hematopoietic stem cells and MSCs [93]. Samara et al. recently suggested that fetal MSCs circulating in the maternal blood may have an immunosuppressive effect on mothers, causing COVID-19 in pregnant women to be often mild and even asymptomatic [94]. Pietras et al. have neatly characterized the cell cycle regulation in hematopoietic stem cells from fetal, adult, and aged sources. According to this characterization, only 0.02% of fetal hematopoietic stem cells are in the quiescent phase (G0), and almost all of them are actively dividing. This is not the case for hematopoietic stem cells in adults and the elderly, where approximately 95% of the hematopoietic stem cells are in the G0 phase [95].

It is evident that the presence of active and proliferating stem cells is one of the main common features in pregnant women and children, a fact that may have a role in the reduced severity of symptoms, hospitalizations, and clinical complications of COVID-19 in these populations.

#### How Do Proliferating Active Stem Cells Resist the SARS-Cov-2 Infection?

Based on the aforementioned differences between quiescent and active stem cells, we herein hypothesize that active stem cells are more successful in resisting the COVID-19 infection. This hypothesis can explain the COVID-19 resistance of children and pregnant women both at the early infection stage and at the time of recovery.

A eukaryotic cell cycle consists of four different stages: gap 1 (G1), synthesis (S), gap 2 (G2), and mitosis (M) [96]. In the G1 phase, the cell gets prepared for the S period (DNA synthesis phase) by synthesizing the macromolecules needed for DNA amplification, such as proteins, saccharides, and RNAs. Following the S period, the cell enters the G2 phase and prepares for cell division (mitosis). The cells that complete the cell cycle are actively proliferating cells. On the other hand, quiescent cells reside in an extended G1 phase, called the G0 phase, where the cells do not divide, and the G0 phase is considered as a phase outside the cell cycle [97]. Checkpoint proteins, including cyclin-dependent kinases (CDKs) and cyclins, play a key role in regulating cell cycle progression [97]. Viruses create a favorable environment for their replication in the host cell by using a variety of molecular strategies, including manipulating the host cell cycle [98]. Coronaviruses use different strategies to exploit host cells for their viral replication [99,100]. For example, avian coronavirus infectious bronchitis virus (IBV) and transmissible gastroenteritis virus (TGEV) reduce cyclin D1, inhibit G2/M transition, and lead to cell cycle arrest at G2/M [101,102]. In contrast, SARS-CoVs are shown to inhibit the G0/G1 progression by using ORF7a and ORF3a. SARS-CoVs reduce the expression of cyclin D3 at both the transcriptional and the translational level by expressing a 7-amino acid protein, thereby reducing Rab phosphorylation. The latter, in turn, prevents cell cycle progression by inhibiting the G0/G1 transition and increases apoptosis [103]. Porcine epidemic diarrhea virus (PEDV) and murine hepatitis virus (MHV) also arrest their host cells in G0/G1 transition [104,105,106]. 

To date, we do not know the exact mechanism of action by which the newly emerged SARS-CoV-2 disturbs the host cell cycle. However, given that the virus is among the SARS-CoV species, it is likely that SARS-CoV-2 arrests its host cell in phase G0. As previously mentioned, it has been shown that SARS-CoV-2 needs the ACE2 receptor to enter the host cell, which explains why the nasal, esophageal, and intestinal cells are the initial target cells of this virus. On the other hand, previous in vitro studies have shown that stem cells from different sources of human tissues have significantly lower expression of the ACE2 receptor. Schäfer R et al. examined MSCs from the three sources of bone marrow, amniotic fluid, and adipose tissue and demonstrated that these MSCs rarely express ACE2 and TMPRSS2, either under normal or inflammatory conditions [22]. In another in vitro study, Hernandez et al. analyzed the expression level of ACE2 and TMPRSS2 on human umbilical cord-derived MSCs from 24 donors and showed that human umbilical cord-derived MSCs express ACE2 and TMPRSS2 at a significantly lower level compared to lung cells [23]. Similar results were reported by Avanzini et al., who investigated fetal and adult MSCs (derived from amniotic membrane, cord blood, cord tissue, bone marrow, and adipose tissue) and confirmed that MSCs derived from different human tissues express ACE2 and TMPRSS2 at very low levels and therefore are not susceptible to SARS-CoV-2 infection [24]. It should, however, be noted that all these in vitro studies used MSCs in their proliferative state (G2/S/M phase) and not in the quiescent state (G0 phase), so they do not reflect the behavior of quiescent stem cells. On a related note, some in vivo studies indicated that COVID-19 significantly reduces the resident stem cell population in the lung tissue [107]. Therefore, it is possible that the quiescent stem cells (G0) resident in the target tissues of adults have enough ACE2 for SARS-CoV-2 to infect them, preventing them from performing their proper function, i.e., the regeneration of damaged cells and the modulation of the immune system. To the best of our knowledge, there are no studies that specifically examine the expression of ACE2 and TMPRSS2 in quiescent stem cells.

The difference in the stem cell composition can explain the smaller rate of infection/hospitalization in infected children/pregnant women versus adults. In other words, it would follow that SARS-CoV-2 infects quiescent stem cells (G0 cells) and hinders their proper function. This hypothesis agrees with the findings regarding the malfunction of resident stem cells located in adult tissues after infection with SARS-CoV-2 [107,108], which subsequently induces an inflammatory response and fibrotic reactions [109]. This is probably why older adults, who have the majority of their stem cell pool in the G0 phase, are more prone to SARS-CoV-2 infection. However, children and pregnant women, who have more active stem cells in their body, show more success against SARS-CoV-2-induced inflammation and more quickly compensate for the damage to injured tissues. 

### 4.2. ACE/ACE2 Ratio

ACE and ACE2 are the main enzymes involved in maintaining the renin–angiotensin system (RAS) homeostasis, regulating blood pressure, and fluid and salt balance. ACE converts angiotensin I (Ang)-I to Ang II, resulting in vasoconstriction. ACE2 is a cell surface enzyme that converts Ang II to Ang I-VII and negatively regulates ACE activity in RAS. The ACE/ACE2 ratio is the clinically determining factor for many pathological conditions. ACE2 is the main receptor recognized by SARS-CoV-2 to enter the host cell. Recently, it has been reported that a higher ACE/ACE2 ratio might increase the risk of severe symptoms in COVID-19 infection [110]. Pagliaro et al. provided a thorough discussion on how the ACE/ACE2 ratio is associated with dramatic COVID-19 complications and lethality. An increased ACE/ACE2 ratio favors in the cell ACE arm in the RAS pathway. This in turn leads to an increased generation of ROS, oxidative stress, inflammation, cytokine storm, alveolar damage, pulmonary hypoxia, and ultimately ARDS. Interestingly, stem cells from aged people show an increased ratio of ACE/ACE2 that seems to be associated with their impaired function in the elderly. Recent studies have shown that the expression of ACE in adult MSCs is significantly higher than in cord blood MSCs [111], which leads to the increased ACE/ACE2 ratio on stem cells from the elderly. Therefore, stem cells from the aged population are not able to properly perform their immunomodulatory and regenerative functions. In contrast, stem cells in children and fetal-derived stem cells in pregnant women have a low ACE/ACE2 ratio, exert their immunomodulatory and regenerative functions, inhibit pulmonary edema hypoxia, and make children/pregnant women more tolerant to COVID-19. This is why active stem cells in children/pregnant women can carry out a protective role against COVID-19 and prevent serious pathological symptoms.

Contrary to the common trend in MSCs, human placenta-derived MSCs (hPMSCs) have been recently reported to express high levels of ACE2 [112]. If hPMSCs are further shown to resist SARS-CoV-2 infection in a similar fashion to MSCs, this could create a scenario in which ACE2 levels and the ACE/ACE2 ratio demonstrate differential contributions to infection tolerance/resistance. More experimental data would elucidate the intricacies of SARS-CoV-2 infection in different tissues.

Furthermore, it is noteworthy to include that neonates (<1 year of age) have increased vulnerability to COVID-19 infection than children. It has been shown that neonates have a higher number of ACE2-positive progenitor cells (resident in their lung) that make them susceptible to infection with SARS-CoV-2 [113]. By comparison, the number of progenitor cells in the lung is significantly decreased in older children. Therefore, the increased vulnerability of infants to COVID-19 might be attributed to the higher number of lung progenitor cells, as well as to their unestablished immune system [113]. However, when compared with adults, infants are still better equipped for recovery from COVID-19, owing to their active stem cells with a low ACE/ACE2 ratio (expectedly, excluding the cases with preexisting conditions such as previous therapeutic regiments, chemotherapy, etc., which compromise the active stem cells).

The idea hypothesized in this work not only can help predict the pattern of spread and severity of the disease in a population, but also can pave the way for the design of novel stem cell-based treatment and prevention strategies for COVID-19. The use of MSCs cell therapy to treat COVID-19 has recently attracted the attention of many researchers. A review of the existing reports suggests satisfactory clinical outcomes in patients treated with MSCs [114,115]. MSCs are known for their potential to modulate the immune system, defend against viral infections, and regenerate tissues [116]. Numerous cellular sources have been used to extract MSCs, with umbilical cord-derived MSCs being the most frequently utilized [14]. Given the promising results of umbilical cord- and Wharton jelly-derived stem cells in the treatment of COVID-19, fetal cellular sources—that seems to have higher numbers of active stem cells—can be speculated to be more reliable than other alternatives. The optimistic outcomes of fetal-derived stem cell therapy for COVID-19 treatment might be due to having a higher proportion of active stem cells and a lower ratio of ACE/ACE2 in fetal-derived MSCs compared to other adult tissue-derived stem cells [111]. Stem cells are also a new platform for designing prophylactic treatments and vaccines [117] due to the secretion of a medium rich in immune-modulating factors [116], inhibitors of pulmonary fibrosis [111], and factors for regenerating damaged cells [19]. Finally, we suggest that a low ACE/ACE2 ratio and cell sorting according to the cell cycle phase, should be additionally considered among the minimum criteria for establishing stem cell-based products for the treatment of COVID-19. Future research and the result of ongoing studies will further elucidate the role of stem cells in warding off SARS-CoV-2 infection and facilitate the development and implementation of stem cell-based therapies for COVID-19.

## 5. Conclusions

The lower rate of death and milder symptoms of COVID-19 infection in children and pregnant women are consistently reported across different regions of the world. Active stem cells (with a low ACE/ACE2 expression ratio) are specific to children and pregnant women. We suggest that in contrast to active stem cells, quiescent stem cells, which are resident in the organs, are attacked by SARS-CoV-2 and lose their proper function. As a consequence, quiescent stem cells are not able to exert their immunomodulatory and regenerative effects in infected patient. In contrast, active stem cells in children and pregnant women can avert SARS-CoV-2, function in immunomodulation, repair damaged tissues, and consequently pave the way for a fast recovery of the infected patient. Furthermore, the unique features of stem cells and the optimistic results of proliferating MSC therapy in patients may provide evidence of a key role of active stem cells in the defense against COVID-19 infection. Given these two observations, we hypothesize that stem cells are the underlying factor in the discriminate COVID-19 infection susceptibility among age groups. This hypothesis can offer a new perspective and lay groundwork for further research on both the prevention and the treatment of the disease. In addition to differences in the activity level and the ACE/ACE2 ratio, other variations among stem cells can be the subject of future investigation for a better understanding of the resistance against COVID-19.

## Figures and Tables

**Table 1 ijms-22-11787-t001:** A summary of reports focusing on the clinical outcomes of COVID-19 in children.

Population under Study	Date of Data Gathering	Number of Patients under Study	Outcome	First Author (Ref)
China	Updated through 11 February 2020	965 (aged ≤19 years)	Only one death occurred in a person aged ≤19 years	Wu Z et al. [10]
China	28 January–26 February 2020 (with follow-up to 8 March 2020)	171 (aged <16 years)	Milder clinical courses and asymptomatic infections were found in children.	Lu X et al. [45]
United States	12 February–2 April 2020	2572 (aged <18 years) out of 149,082 COVID-19 patients	Three deaths were reported in this analysis. Most of the hospitalizations occurred in infants (aged <1 year). Whereas severe cases of COVID-19 are rare in children, serious illness resulting in hospitalization still occurs.	[46]
China	From 17 January to 1 March 2020	36 (aged <16 years)	All pediatric patients manifested mild or moderate symptoms of infection.	Qiu H et al. [47]
China	From 1 February till 10 February 2020	35 (aged from 1 month to 14 years)	Children were reported to be as sensitive to COVID-19 as adults, while clinical outcomes were more favorable in children. Children under 3 years of age had the highest risk of developing serious illness and need more intensive medical care than other children.	Zheng F et al. [51]
China	from 16 January to 6 February 2020	15 (aged from 4 to 14 years)	Small nodular ground-glass opacities were the main findings of the early chest CT images of children with 2019-nCoV infection.	Feng K et al. [52]
China	From 24 January to 24 February	8 severe cases (aged from 2 month to 15 years)	Common symptoms were polypnea, fever, cough, and cytokine storm. Imaging changes were multiple patch-like shadows and ground-glass opacity.	Sun D et al. [53]
North America	14 March–3 April 2020 (with follow-up to 10 April 2020)	48 (aged 4.2–16.6 years)	Severe illness was far less frequent in children than in adults.	Shekerdemian LS et al. [54]
New York City	Over a 1-week period in late March 2020	2 infants	Clinical course was benign in both infants.	Paret M et al. [55]
Malaysia	until end of February 2020	4 (aged 1.7–11 years)	Mild or asymptomatic presentation in children was reported.	See KC et al. [56]
Washington, DC	15 March–30 April 2020	177 (children and young adults)	The risk of being hospitalized was higher in the youngest (<1 year) and oldest children/young adults (15–25 years of age).	DeBiasi RL et al. [57]
China	January–February 2020	33 infants (born to mothers with COVID-19, including 3 neonates with COVID-19)	Mild clinical symptoms were seen in 33 neonates with or at risk of COVID-19. Outcomes were favorable.	Zeng L et al. [58]
Nationwide case series (reported to the Chinese Center for Disease Control and Prevention)	16 January–8 February 2020	2135 (aged 2–13 years)	More than 90% of all patients were asymptomatic, mild, or moderate cases. Young children, particularly infants, were vulnerable to infection.	Dong Y et al. [59]
Italy	3 March–27 March	100 (aged <18 years)	Most of the infants presented with mild disease. Severe and critical cases were diagnosed in patients with (10) conditions. No deaths were reported.	Parri N et al. [60]

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
