# Peer review of "A Review on the Role of Stem Cells against SARS-CoV-2 in Children and Pregnant Women"

_ijms, 2021, doi:10.3390/ijms222111787_

Round 1

Reviewer 1 Report

In this review paper, the authors provided a review of recent reports on the unique clinical outcomes of COVID-19 in children and pregnant women, who have a relatively low incidence and severity of the disease. They then proposed a hypothesis that pregnant women and children are more tolerant to SARS-Cov-2 infection due to the pool of actively proliferating stem cells with a very low expression level of ACE2 and a lower ratio of ACE/ACE2. Overall, this is an interesting review that has highlighted a low expression level of ACE2 and a lower ratio of ACE/ACE2 as a potential molecular signature for COVID-19 in pregnant women and children. However, to make this review more appealing, the following points need to be addressed.

  • As stated in this review, several lines of evidence have shown that adult MSCs cultured in vitro are negative or express a very low level of ACE2 (Ref. 22, 23, 24), supporting the notion that actively proliferating stem cells express a lower level of ACE2 and are resistant to SARS-Cov-2 infection. However, a most recent study indicate that human placenta-derived MSCs (hPMSCs) express a high level of ACE2, which contribute to their protective effects on ischemic stroke (Barzegar M et al. Stem Cells, 2021). Due to the perinatal origin, hPMSC should have a relatively high proliferation activity but still express ACE2. Therefore, it is necessary to provide more discussions on this controversy.
  • On page 5, line 163-164, hASCs should be highlighted in the parentheses since more details about hASCs are presented in the following section of the text.
  • Page 5, line 203, “the COVID-19 infection” should be replaced by “SARS-Cov-2 infection”.
  • Since a subtitle “4.1.1. How do proliferation active stem cells resist the COVID-19 infection?” is given (line 203), is a subsequent subtitle (4.1.2.) missed on page 6, line 227-228?

Author Response

Dear Editor,

I appreciate your feedback and the reviewers’ constructive comments. Below, we provide a point-to-point response to the comments. We hope you find these revisions satisfactory. We are ready to make further amendments if necessary.

Sincerely,

Rahim Raoofi Jahromi, MD

The corresponding author

Reviewer 1

In this review paper, the authors provided a review of recent reports on the unique clinical outcomes of COVID-19 in children and pregnant women, who have a relatively low incidence and severity of the disease. They then proposed a hypothesis that pregnant women and children are more tolerant to SARS-Cov-2 infection due to the pool of actively proliferating stem cells with a very low expression level of ACE2 and a lower ratio of ACE/ACE2. Overall, this is an interesting review that has highlighted a low expression level of ACE2 and a lower ratio of ACE/ACE2 as a potential molecular signature for COVID-19 in pregnant women and children. However, to make this review more appealing, the following points need to be addressed.

We would like to express our sincere thanks to you for the positive comments. Based on your comments and suggestions, we have made relevant modifications to the manuscript.

Q: As stated in this review, several lines of evidence have shown that adult MSCs cultured in vitro are negative or express a very low level of ACE2 (Ref. 22, 23, 24), supporting the notion that actively proliferating stem cells express a lower level of ACE2 and are resistant to SARS-Cov-2 infection. However, a most recent study indicates that human placenta-derived MSCs (hPMSCs) express a high level of ACE2, which contributes to their protective effects on ischemic stroke (Barzegar M et al. Stem Cells, 2021). Due to the perinatal origin, hPMSCs should have a relatively high proliferation activity but still express ACE2. Therefore, it is necessary to provide more discussions on this controversy.

R: Contrary to the common trend in MSCs, human placenta-derived MSCs (hPMSCs) are recently reported to express high levels of ACE2 (Barzegar et al., 2021). If hPMSCs are further shown to resist SARS-CoV-2 infection in a similar fashion to MSCs, it could be one of the scenarios in which ACE2 levels and ACE/ACE2 demonstrate differential contributions to the infection tolerance/resistance. More experimental data would elucidate the intricacies of SARS-CoV-2 infection in different tissues.

*The explanation and the related reference were included in the manuscript (section 4.2. ACE/ACE2 ratio).

On page 5, line 163-164, hASCs should be highlighted in the parentheses since more details about hASCs are presented in the following section of the text.

It was revised as suggested.

Page 5, line 203, “the COVID-19 infection” should be replaced by “SARS-Cov-2 infection”.

It was revised as suggested.

Since a subtitle “4.1.1. How do proliferation active stem cells resist the COVID-19 infection?” is given (line 203), is a subsequent subtitle (4.1.2.) missed on page 6, line 227-228?

There is no missed subtitle in this line.

Reviewer 2 Report

This is a nicely written review that discusses the mechanisms behind low COVID-19 infectivity in children and pregnant women. The authors suggest that low expression of angiotensin-converting enzyme 2, and the lower ratio of ACE/ACE2 of stem cells in children and pregnant women may be the reason behind mild symptoms of COVID-19. The review also touches on how stem cells exert immunomodulatory potential in COVID-19 infected children and pregnant women. The study also provides potential stem cell-based therapeutics for COVID-19. The review could be improved to deeply cover some aspects. Please discuss the molecular determinants of juxtacrine and paracrine stem cells/MSC-based improvement of respiratory disease. Please discuss available literature about interaction between stem cells/MSC and lung Th1, Th17 cells, DCs, and alveolar M1 macrophages. Please discuss production of TGF-β and its effect on T cell cycle progression. Please also discuss the angiomodulatory factors produced by MSC that improve oxygen supply. The paragraph discussing the therapeutic potential of stem cells/MSC in treating COVID 19 is currently superficial. Please discuss the available experimental evidence and clinical trials of MSC-based treatment of COVID 19/respiratory disease.

In addition, the manuscript in its current form suggests that stem cells may be the only protective factor in pregnant women which is misleading.

It is well known that progesterone is an important immunomodulatory and anti-inflammatory hormone produced by the placenta. It has been shown that progesterone exerts anti SARS-CoV-2 activity (Gordon DE, Nature, 2020). I suggest the authors to rationalize why they focus on stem cells but not hormones as well as protective factors during pregnancy. Please add during which trimester pregnant women are mostly protected from the outcomes of SARS-CoV-2 infection. A paragraph to discuss progesterone receptor expression in immune cells/MSC as well as progesterone-induced skewing of CD4+ T-helper cell from Th1 to Th2, production of anti-inflammatory cytokines and the increase in FOXP3+ Treg cells leading to immune tolerance during pregnancy will improve the current content. In addition, estrogen and progesterone stimulate antibody production by B cells which is related to SARS-CoV-2 infection. Please refer to the clinical trials NCT04359329 and NCT04365127 for more information.

Please refer to Leng, Zikuan, et al. Transplantation of ACE2-mesenchymal stem cells improves the outcome of patients with COVID-19 pneumonia, Aging and disease, 2020.

Minor comments

Line 129: all of whom

Line 136: Please change zero per cent to number and symbol, through out the manuscript

Line 212: are actively proliferating

Line 246: proliferative state

Author Response

Dear Editor,

I appreciate your feedback and the reviewers’ constructive comments. Below, we provide a point-to-point response to the comments. We hope you find these revisions satisfactory. We are ready to make further amendments if necessary.

Sincerely,

Rahim Raoofi Jahromi, MD

The corresponding author

Reviewer 2

This is a nicely written review that discusses the mechanisms behind low COVID-19 infectivity in children and pregnant women. The authors suggest that low expression of angiotensin-converting enzyme 2, and the lower ratio of ACE/ACE2 of stem cells in children and pregnant women may be the reason behind mild symptoms of COVID-19. The review also touches on how stem cells exert immunomodulatory potential in COVID-19 infected children and pregnant women. The study also provides potential stem cell-based therapeutics for COVID-19. The review could be improved to deeply cover some aspects.

Q: Please discuss the molecular determinants of juxtacrine and paracrine stem cells/MSC-based improvement of respiratory disease. Please discuss available literature about interaction between stem cells/MSC and lung Th1, Th17 cells, DCs, and alveolar M1 macrophages. Please discuss production of TGF-β and its effect on T cell cycle progression. Please also discuss the angiomodulatory factors produced by MSC that improve oxygen supply. The paragraph discussing the therapeutic potential of stem cells/MSC in treating COVID 19 is currently superficial. Please discuss the available experimental evidence and clinical trials of MSC-based treatment of COVID 19/respiratory disease. In addition, the manuscript in its current form suggests that stem cells may be the only protective factor in pregnant women which is misleading. It is well known that progesterone is an important immunomodulatory and anti-inflammatory hormone produced by the placenta. It has been shown that progesterone exerts anti SARS-CoV-2 activity (Gordon DE, Nature, 2020). I suggest the authors to rationalize why they focus on stem cells but not hormones as well as protective factors during pregnancy. Please add during which trimester pregnant women are mostly protected from the outcomes of SARS-CoV-2 infection. A paragraph to discuss progesterone receptor expression in immune cells/MSC as well as progesterone-induced skewing of CD4+ T-helper cell from Th1 to Th2, production of anti-inflammatory cytokines and the increase in FOXP3+ Treg cells leading to immune tolerance during pregnancy will improve the current content. In addition, estrogen and progesterone stimulate antibody production by B cells which is related to SARS-CoV-2 infection. Please refer to the clinical trials NCT04359329 and NCT04365127 for more information. Please refer to Leng, Zikuan, et al. Transplantation of ACE2-mesenchymal stem cells improves the outcome of patients with COVID-19 pneumonia, Aging and disease, 2020.

We are grateful for the encouraging feedback. To address the reviewer’s comments, we included the requested details and the related references in the text.

R: Positive therapeutic results have been reported when utilizing MSCs in the treatment of respiratory complications (Laffey & Matthay, 2017; Lanzoni et al., 2021; Leng et al., 2020; Matthay, 2015; Matthay, Goolaerts, Howard, & Lee, 2010). MSCs are thought to use both juxtacrine (i.e. direct cell-cell interaction) and paracrine (secretion of extracellular vesicles and exosomes) paths (Carl Randall Harrell, Jovicic, Djonov, & Volarevic, 2020). MSCs help regenerate tissues and reduce cell-based and humoral immune responses through immunomodulating factors such as FGF, HGF, EGF, VEGF (Obendorf, Fabian, Thome, & Laube, 2020). For instance, secretion of HFG by MSCs helps to induce dendritic cell immune tolerance and ease acute lung injury (ALI) (Z. Lu et al., 2019). MSCs are shown to reduce the inflammation of the epithelial lung cells by transporting miR21-5p and increasing the polarization of alveolar macrophages from the pro-inflammatory to anti-inflammatory phenotype (wei Li, Wei, Han, & Chen, 2019). Pro-inflammatory factors made by M1 macrophages at the site of injury activate resting MSCs, whose induced immunosuppressive characteristics, in turn, lower the overall immune response (Wang et al., 2021). MSCs further suppress the proliferation of the activity of lung Th1 and Th17 helper T lymphocytes through a plethora of secreted cytokines including TGF-β and HGF (Rawat, Gupta, & Mohanty, 2019). MSCs similarly suppress the cell cycle of B lymphocytes, resulting in a diminished expression of a series of chemokine receptors (such as CXCR4 and CXCR5) and immunoglobulins (IgM, IgG, IgA) (L. Fan et al., 2016). Alongside the anti-inflammatory actions of MSCs, they further facilitate the restoration of the damaged tissues from ARDS, ALI, and other repertory inflammatory complications by their angiomodulation (C Randall Harrell et al., 2019). Their support for the generation of blood vessels, through the production of a spectrum of proangiogenic factors such as VEGF, HGF, TGF-β, MCP-1, angiopoietin-1, etc., reinforces the recovery process of injured pulmonary tissues by addressing the oxygen supply (Maacha et al., 2020).

*The explanation and the related reference were included in the manuscript (section 1. Introduction).

Early studies do not indicate that there is a drastic difference between the levels of infection resistance in pregnant women in different trimesters; However, more experimental data might be needed to confidently draw a conclusion on the subject (Rajewska et al., 2020; Rosen et al., 2021; C.-L. Wang, 2021). The intrinsic immune tolerance of pregnancy in general, and all or part of its underlining molecular mechanism, including the roles of hormones can contribute to the curious resistance of pregnant women to COVID-19 (Gordon et al., 2020; Lissauer et al., 2015; Zhao et al., 2012); However, alongside reviewing COVID in children and pregnant women, the authors tried to highlight common factors between these two demographic groups which may explain COVID tolerance in these populations simultaneously.

*The explanation and the related reference were included in the manuscript (section 3. COVID-19 in Pregnant Women).

Minor comments

Line 129: all of whom

Line 136: Please change zero per cent to number and symbol, throughout the manuscript

Line 212: are actively proliferating

Line 246: proliferative state

* All revised as suggested.